# The child Musicality Index: A child-friendly version of the Goldsmiths Musical Sophistication Index

Chloe MacGregor[1]*, Solena Mednicoff[2], David J. Vollweiler[2], Erin Hannon[2], Daniel Müllensiefen[1]

1 Department of Psychology, Goldsmiths, University of London, London, England, United Kingdom,
2 Department of Psychology, University of Nevada, Las Vegas, Nevada, United States of America

* cstac001@gold.ac.uk

## Abstract

The Goldsmiths Musical Sophistication Index (Gold-MSI) has contributed significantly to the study of musical sophistication in adolescence and adulthood, however the field lacks an equivalent self-report instrument for childhood. Existing measures focus heavily on instrumental learning and fail to consider social, emotional and motivational factors which are likely to underlie engagement with music. To address this gap, the current research introduces the child Musicality Index (cMI), a new psychometric self-report measure of musicality for 6- to 13-year-olds which captures children's motivation for and enjoyment of music. In study 1, a large cohort (N = 302) of children (6–13yrs) responded to items adapted from the Gold-MSI to suit a younger age range. Factor analysis (N = 283) was used to select 8 items for the children's version of the measure, split into two group factors ('musical drive' and 'enjoyment of music making') and one general factor ('musicality'). Studies 1b and 1c then investigated whether children's responses to these 8 items were valid and reliable (N = 250–460). Study 2 provided evidence for the psychometric properties of the final 8-item version of the task in a new sample (N = 56). The new, 8-item cMI provides a short and robust measure which can facilitate investigation of the most critical aspects of musicality during childhood. It has demonstrated good psychometric properties in UK and US samples and is now openly available for use in wider research.

## Introduction

The Goldsmiths Musical Sophistication Index (Gold-MSI [1]) is a popular psychometric tool that has been used in a great number of research projects since it's development. It uniquely offers a standardised assessment of musical expertise and skilled behaviour that covers a diverse range of musical abilities, is appropriate for measuring individuals at all levels of musical experience and has been translated into several languages. In providing the research community with an effective and versatile

**Data availability statement:** The data will be held in an OSF repository (https://osf.io/jgrk6/overview?view_only=bf376f77b1d-945d8aa32f1bbed7df4f2) which will be made public upon publication.

**Funding:** This work was supported by a grant from Misophonia Research Fund to EH.

**Competing interests:** The authors have declared that no competing interests exist.

tool, the Gold-MSI has allowed for significant developments in the study of musicality in adolescence (i.e., from 10 to 20 years of age; see the LongGold project [2,3]) and adulthood [4–8]. However, despite the advances attributed to the Gold-MSI in adolescent and adult populations, there is no equivalent instrument for the investigation of musicality in childhood.

A new psychometric scale suitable for the quantitative assessment of child musicality would be of benefit to the field of musical development. Existing questionnaires only provide qualitative results or fail to account for the rich and varied nature of musical interactions during childhood by focusing instrumental training. Alternative methods of measuring child musicality, such as gathering reports from parent and teachers, have been called upon to address this problem. However these methods are subject to informant bias and do not account for facets of musical motivation, communication and understanding which are highly relevant to music in childhood [9]. To address this gap, we introduce a new child-friendly self-report scale suitable for measuring musicality in 6–13-year-olds. The aim is to ensure researchers in the field of musical development are equipped with a tool suitable for measuring musicality as it manifests in childhood.

## Musical sophistication and musicality

Musicality in adulthood has been described as an individuals' ability to engage with music in a flexible, effective and refined manner [1]. The Gold-MSI was built to capture the wide range of skills and behaviours that are associated with 'musical sophistication'. It includes five separate facets: active engagement, perceptual abilities, musical training, singing abilities and emotions. Aside from the musical training component, these facets are neither defined by nor depend wholly on formal training and vary throughout the adult population [7].

Child musicality can be differentiated from musical sophistication in adulthood [10]. As a child, you are more likely to experience adult-selected music that accompanies play or learn music as part of an adult-led activity, whereas during adulthood, you are more likely to instigate musical activities with others as a method of communicating and creating shared experience, for example [11]. This reflects a broader distinction between childhood and adulthood regarding the extent to which individuals can influence their musical environment. Generally, young children do not have many opportunities to pursue their own interests. With age, however, autonomy increases, and they can choose whether they engage with musical interests. Underlying this process is a shift from the importance of the 'passive genotype', where experiences are more influenced by features of the rearing environment that correspond to the child's and parent's shared genetics, to a greater importance of the 'active genotype', which allows individuals to seek out environments consistent with their personality, skills, and interests, especially as they get older and more autonomous [12]. Moving to adulthood, an increase in autonomy inevitably influences the extent of musical engagement to a greater degree than it does during childhood.

Musical training is an example of a facet of adult musical sophistication which may relate differently to musicality during childhood. Whereas adults may choose

to pursue music lessons in their free time, music education is part of the compulsory school curriculum in most Western societies [13] and parents have power to dictate children's engagement with training activities, especially when children are young [14]. In addition, music activities in and outside of a child's school could exhibit greater variation than would be typical of adult music training. Children's music activities could vary from attending community music classes with their families ("Mommy & Me" classes, for example) to taking formal piano lessons. It is therefore important to conceptualise musicality in childhood independently from musical sophistication in adulthood and approach its measurement with this in mind.

## Measuring child musicality

Researchers have taken various approaches to the objective measurement of child musicality. One common approach involves measuring select observable musical skills and using these to infer overall musicality. Two measures frequently employed in child samples are Gordon's Primary Measures of Music Audiation [15] and Seashore's Measure of Musical Talent [16], which involve simple perceptual discrimination tasks of pitch and rhythm. These tests are popular with researchers because they are short and easy to complete with large groups of children, and they provide quick, objective scores of children's ability. However, they cannot fully engage the natural mechanisms underlying musicality as a result of being oversimplified and far removed from typical musical experiences during childhood. It is problematic that scores from perceptual tasks such as these are commonly misinterpreted as measures of musicality. While perceptual abilities can contribute to child musicality, there are many other characteristics and abilities such as motivation, musical communication and musical understanding which are more difficult to measure objectively but also play a significant role [9]. The rich musical lives of children should therefore not be reduced to outcomes on these simple perceptual tasks.

Another approach asks informants, usually parents and teachers, to comment on a child's musical skills using standardised questionnaires. Questionnaires overcome the practical constraints of ability testing and can consequently facilitate investigation of a wider range of skills. One questionnaire [17] asks teachers to provide ratings of children's singing ability, their recognition of songs, tunes and timbres and their responses to music and rhythm. This measure thus accounts for children's musical production skills and their typical engagement with music on top of providing an estimate of perceptual skills. Similarly, another questionnaire [18] asks parents to rate their children's interest in music, emotional responses to music, musical creativity and musical training, as well as report when children first started to manifest particular musical behaviours. On top of perception and production skills, this measure includes estimates of emotional response and musical creativity. Overall, these informant measures can provide a much more detailed account of child musicality than perceptual tasks alone.

Despite these advantages, questionnaires have been adopted in child musicality research less often than perceptual discrimination tasks. This could be because collecting data from parents and teachers is typically dependent upon strong communication between researchers, schools and parents, and can be affected by time limitations. This may be especially true for teachers in the case that they must complete questionnaires for many children in a class. Given that psychometric properties of informant questionnaires are rarely measured and reported, it could also be the case that researchers prefer to ensure they are gathering a robust estimate of musicality in childhood by using a more objective perceptual measure that has been shown to be reliable and valid. One recently developed instrument, the Child Musicality Screening (CHIMUS [19]), provides researchers with an informant questionnaire that is both short and reliable ($a = .86$). This represents a positive shift in the current field towards measuring children's musicality in greater depth.

Unfortunately, however, the accuracy of parent and teacher reports is difficult to ensure. Accuracy depends on adults having insight into children's musical lives and may vary depending on the informants' level of contact with children. It is possible that parents do not frequently witness their children being musical, particularly if they do not have a lot of time to spend with them or do not naturally engage in musical interactions at home [20], for example. Teachers typically engage with children in class groups rather than on an individual basis, meaning in some cases they may not be able to provide a

sufficiently detailed or individualised account. Conversely, the accuracy of informant reports should also be questioned in the case that parents and teachers rate children they spend a lot of time with and know very well. Parents and teachers naturally have vested interests in children and could therefore be tempted to exaggerate their ability ratings to present them in a favourable light. On top of this, and regardless of time spent being musical with children, parents and teachers cannot gain a complete insight into the subjective experiences of children and given the complex nature of musicality, especially considering the influence of various social, emotional and motivational factors, this may be especially challenging. Researchers must therefore be wary of potential bias and lack of insight when relying on informant reports.

One possible solution to these issues could be validating parent and teacher ratings against children's self-reports [21]. The Motivation for Learning Music questionnaire (MLM; [22]) adopts this approach with a two-part scale completed by parents and children. Overall, the scale demonstrates good reliability and convergent validity, demonstrating the benefit of including children's self-ratings alongside those of parents or teachers in assessment of music-related constructs in childhood. However, the Motivation for Learning Music questionnaire focuses on assessing motivation for formal training in piano or violin and is therefore unable to assess children who do not choose to learn an instrument or do not have access to formal training. Further, it does not account for everyday forms of musical engagement, such as singing and dancing, which are vital to musicality during childhood [23,24].

One child-report which takes an inclusive approach and provides a more well-rounded view of musicality is the Pupils' Music Questionnaire [25]. This instrument investigates children's attitudes to engagement with music in and outside of school. It asks about instrument learning but also accounts for music listening and music making and can therefore be administered to children with no musical training experience. Unfortunately the Pupil's Music Questionnaire, similar to other measures that can assess children without training (e.g., the Child Music Interest Interview [26]), does not offer a numerical estimate of musicality making it unsuitable for use in quantitative research. Researchers who wish to conduct a quantitative comparison between children are thus faced with a lack of appropriate instruments.

To our knowledge, there has yet to be a tool developed which is suitable for children regardless of musical training, can capture a reliable quantitative estimate of child musicality and can account for aspects of musicality that are most relevant during childhood. The current paper therefore describes the development of a new psychometric scale to capture musicality in children aged 6–13 years. In line with well-established scales such as the MLM [22], we aimed to establish a scale that can be used across a wide age range to capture meaningful variation in musicality across middle childhood.

Two studies were carried out to develop and validate the scale. The first aimed to select a set of 5–15 items to be included in a short scale and assess the validity and reliability of children's responses to these items. The second aimed to validate the item set in a new sample. Items adapted from the original Gold-MSI [1] and the Musical Child Questionnaire (MCQ [9]) were used as a starting point to account for all musical behaviours, including those which might only be relevant during childhood.

## Study 1a

The first study investigated the suitability of items adapted from the original Gold-MSI and the MCQ for a new self-report measure of child musicality. We aimed to identify a maximum of 5–15 items that could be included in a short scale.

### Method

**Participants.** Data were collected from a total of 302 English speaking children with written parental consent (145 female, 151 male, 1 non-binary and 5 who did not provide demographic information) in Las Vegas, US and London, UK between 3/1/2022 and 01/09/2024. They were aged 6–13 years ($M=9$, $SD=2.23$, $N=298$). There were 189 children who took part in Las Vegas ($M=9.35$, $SD=2.35$; 92 female, 96 male, 1 non-binary) and 113 children in London ($M=8.41$, $SD=1.89$; 53 female, 55 male and 5 with no demographic data). The study was independently approved by the ethics committees at the University of Nevada, Las Vegas (UNLV) and Goldsmiths, University of London.

**Materials.** All 38 items from the original Gold-MSI [1] plus 3 items gathering information about learning instruments and absolute pitch were submitted to an iterative rewording process to make them suitable for ages 6–13. For each item, four of the authors indicated their confidence that a 6- to 9-year-old would understand each statement on a scale of 1–5 (with 5 being most confident). For items with ratings of 3 or lower each author was asked to provide a suggested rewording. Rewordings entailed using smaller and more common words, shorter sentences, using positive instead of negative wording for some items (e.g., reverse coded items), and in some cases, changes to content. For example, one item that referred to disposable income was revised to assess the relative value of musical activities compared with other high-value activities to children such as video games, TV, or playing outside. In addition to these 41 items, we also included 7 items from the MCQ [9] which did not require adaptation for use with children. See spreadsheets S1 and S2 Data in supporting information for a full outline of original and edited items.

Readability and psycholinguistic tools were then used to ensure item statements corresponded to average reading age levels. The final statements ranged from 3 to 4 years of age according to the N-Watch (an average of 266.6), which considers age-of-acquisition norms [27], and 8–9 in average reading age, or grade 3, according to the ATOS (4.67 level), which assesses overall text complexity and readability. This use of readability tools is consistent with other studies in the literature that reduce linguistic complexity from adult- to child-appropriate levels for child-friendly diagnostic questionnaires [e.g., 28].

A 5-point Likert scale was chosen in place of the 7-point scale used in the original Gold-MSI to make the task easier for young children. Based on previous reports that children of 5–12 years of age respond in a similar manner to both 3- and 5-point scales [29], we opted for a 5- instead of 3-point scale to preserve the discriminatory power of items. An agreement scale was suitable for most items: strongly disagree "1"- strongly agree "5". This scale was chosen to replicate the format of items in the original Gold-MSI, however it is important to note that the use of agreement scales has previously been validated for research with children 6–13 years of age [30]. Item-specific response options were written for 6 items where an agreement scale was not suitable. In these cases, a 5-point Likert scale was maintained for consistency.

**Procedure.** While all children responded to the same 48-item measure, the research procedure differed by lab. Data collection with children recruited from Las Vegas took place in person, while their parents answered a survey online prior to participation. Children completed the study as part of a larger 1-hour, two visit battery of assessments held within the UNLV Music Lab. At the beginning of the session, children were informed that they would be taking part in a research project and were asked to give assent before participation. For ethics purposes, they were also informed that they did not have to participate and that they could withdraw from the study for any reason by letting the experimenter know. All participants were tested individually. Participants aged 6–9 years had their instructions read aloud ("please answer the following questions with how much you agree with them about yourself") and recorded by the experimenter, and participants aged 10–13 years took the study on their own, with an experimenter monitoring in the same room and able to answer any questions. The 48-item measure was administered via Qualtrics (Qualtrics, Provo, UT) on an iMac computer (Apple, Inc., Cupertino, CA) which was booted into Windows 7 (Microsoft Corporation, Seattle, WA).

Data collection with children from London took place either in person or online. In person, children from two primary schools in East London completed the study as part of a 45-minute research session held in their music classroom or school library. Test sessions were held with between 5 and 8 children in the same school year group at a time; younger children were tested in smaller groups to ensure they could access additional help if required.

At the beginning of the session children were informed that they would be taking part in a research project. For ethics purposes, they were also informed that participation was not compulsory, that there would be no repercussions for withdrawing from the study and that they should inform experimenters if they wanted to stop participating by raising their hand.

The questionnaire was presented on paper. The experimenter read out the first item from the questionnaire and explained how to select an appropriate response by providing examples. Children were then given an opportunity to ask questions before reading and responding to the items independently. Each child completed a set of 24 items from the

48-item set due to time limitations. The 24-item version of the questionnaire had been pre-specified ahead of sessions with items selected at random. Children took around 10 minutes to complete a 24-item questionnaire.

Online participants were recruited by sending parents a link via their child's school. The link took them to a Qualtrics survey which first directed them to complete the parental consent form and fill out age, gender and language demographics for their child. They then received the following instructions:

"The rest of the survey is for your child to complete. Please let them answer themselves and help them if they do not understand a question. Children do not have to complete all questions at once. This survey will automatically save the responses so far and you can come back later by following the same link if they need a break. Please make them aware that they do not have to answer these questions, and they may stop taking part at any time if they no longer wish to participate."

Parents next asked children to complete the remainder of the task. Children were presented with a graphic of a smiley face with headphones and told they would be answering some questions about music. They were instructed to select a response that best described them. Children were then presented with 48 items in 5 blocks of 9 or 10 items at a time. After each block, children were presented with a pie chart informing them of their progress. Following this, children and parents were thanked and provided with a debrief which included information on how to remove their child's data from the study should they have wished to.

**Analyses.** Exploratory and confirmatory factor analytic approaches were used to investigate the structure of children's responses to the 48-item questionnaire. Confirmatory factor analysis is typically used to assess whether a pre-specified factor structure fits a new set of data, while exploratory analysis is used to identify new factors based on correlations between item responses. Confirmatory factor analysis was applied first, using the *cfa* function from the lavaan package in R [31], to rule out the factor structure of the original Gold-MSI as a potential model for the data collected from children. Then a series of exploratory factor analyses were carried out to examine alternative factor models for explaining children's responses, comparing between those with different underlying structures to see which provided the best fit.

An iterative approach was taken, which had been adopted in the development of similar self-report scales in the past [32,33]. It involved computing a hierarchical omega coefficient, which provides an estimate of how well a test is able to capture a single, underlying construct (>.6 suggests the presence of a general factor [34]). We used the *omega* function from the psych package in R [35] to compute this estimate.

Following this, exploratory factor analyses were used to compare a model with one general factor (or not, if none was present), to models with varying numbers of additional factors (2 factors + 1 general factor, 3 + 1, 4 + 1…) After selecting the best fitting model from the alternatives, according to the Bayesian Information Criterion (BIC), items which were highly correlated with the factors in the model were retained and the factor structure of the retained items was assessed. If the number of recommended factors in the best-fitting model remained the same, then this model was recommended as the final solution. If not, the reduced set of items were entered into a further set of analyses, where models with varying numbers of additional factors were compared once more. This procedure was repeated until a stable factor solution was identified where all items had a substantial loading on only one group factor in addition to the general factors and no further changes in the number of items or number of factors were suggested according to the BIC. The R script used for the current analyses have been made available for researchers interested in replicating this approach [36].

These analyses were carried out to reduce the number of items based on a well-fitting model which could explain the pattern in children's responses to the larger 48-item set, eliminating those which did not correlate highly with underlying factors and retaining those which contributed the most explanatory power. We aimed to identify a maximum of 5–15 items to be included in a short scale for the assessment of children's musicality.

## Results

Children's responses to 48-item scale (N = 302) were entered into an iterative set of exploratory factor analyses. We input 45 items which had responses on a 5-point Likert scale, excluding 3 items which asked about perfect pitch and

instrument learning. Data from 15 children who had provided no responses to any items were also excluded, leaving a sample of N = 287. Any remaining cases with missing responses were considered missing at random (MAR) and were thus accounted for using the full information maximum likelihood method (FIML).

Item response distributions were checked for each of the 45 items. All items demonstrated skewness and kurtosis levels of <1.5 and were subsequently retained.

To rule out the factor structure of the original Gold-MSI as a suitable model for children, we carried out a confirmatory factor analysis run using the *cfa* function from the lavaan package in R [31]. The resulting model indicated acceptable fit according to RMSEA (.051) and SRMR (.071) but not according to CFI (.75) and TLI (.72).

An exploratory analysis was then carried out. We found a hierarchical omega coefficient of.7, indicating the presence of a general factor (>.6) [34]. To ascertain how many factors were necessary in addition to a general factor, a series of bi-factor solutions were computed using minimum residual factoring with oblimin rotation. Four bi-factor models including a general factor as well as 3–6 group factors each were output and compared with two additional models with 2 + 1 and 1 factor structure.

Model comparison of BIC values indicated that a 4 + 1 model provided the best fit. On inspection, we identified 33 items with communalities of <.4 indicating they could not be well explained by an underlying factor. We subsequently recalculated the bi-factor models retaining 12 items with high communalities (>.4) and compared them to check whether 4 group factors were still optimal. This revealed that the structure with two group factors had the lowest BIC so we repeated the process, this time retaining only 8 items with high communalities. A third model comparison indicated that this 2 + 1 model with 8 items was optimal, with a substantial difference of 52.57 between the BIC of this and the next best fitting model. Further inspection revealed 2 out of 8 items loaded onto both group factors at a ratio close to 1.5. As these two items could not be assigned to an individual group factor without ambiguity we decided to retain them as part of the general factor, named 'general musicality', along with three items in each of the two group factors (see Fig 1 and Table 1).

The first group factor was labelled 'Musical Drive' to summarise three items that addressed how much children are driven to engage with and prioritise music, e.g., 'Music is one of my favourite hobbies'. The second factor was named 'Music Making' as it comprised three items which reflect children's enjoyment of music making in different contexts, e.g., 'Making up music is a lot of fun for me'. Fig 1 and Table 1 summarise the factor structure and include item statements and factor loadings.

The final 2 + 1 solution with 8 items (see Table 1) was based on data from 283 participants with no missing data patterns. Model fit indices demonstrated that the factorial structure fit this sample well: CFI = .991; TLI = .983; RMSEA = .051; SRMR = .035. Model invariance tests established metric and scalar invariance with respect to gender of child (N = 224: 145 female, 151 male): CFI < .01; RMSEA < .01. Some age groups did not respond to all 8 items so we were unable to carry out age invariance tests.

'Good' reliability of the new general and group factors in the current sample was indicated by Cronbach's alpha and MacDonald's omega values > .71 (see Table 2). The size of sub-samples used for this reliability analysis differed between factors, however a sufficient sample of at least N = 225 completed every item for each factor.

The final 8 items had age-of-acquisition norms that ranged from 3 to 4 years of age, with an average score of 232.6 calculated using the N-watch. Readability according to the ATOS calculated the level to be at 3.62, meaning that the final 8 items are at about a grade 2 reading level, or between 7–8 years in average reading age. While this indicated that 6- and 7-year-olds might have difficulty understanding these items, we wanted to ensure adapted items were still comparable to the original Gold-MSI items and thus decided against simplifying the items further. To assess whether the discrepancy between average reading age of items and the age of our sample would affect test validity we analysed children's response patterns. This involved estimating random and 'extreme' biases, characterised by inconsistent responses or frequent use of the extreme ends of a Likert scale, which could be caused by a lack of understanding (see S3 File for further detail). According to our estimates, 6- & 7-year-olds experienced more difficulty understanding

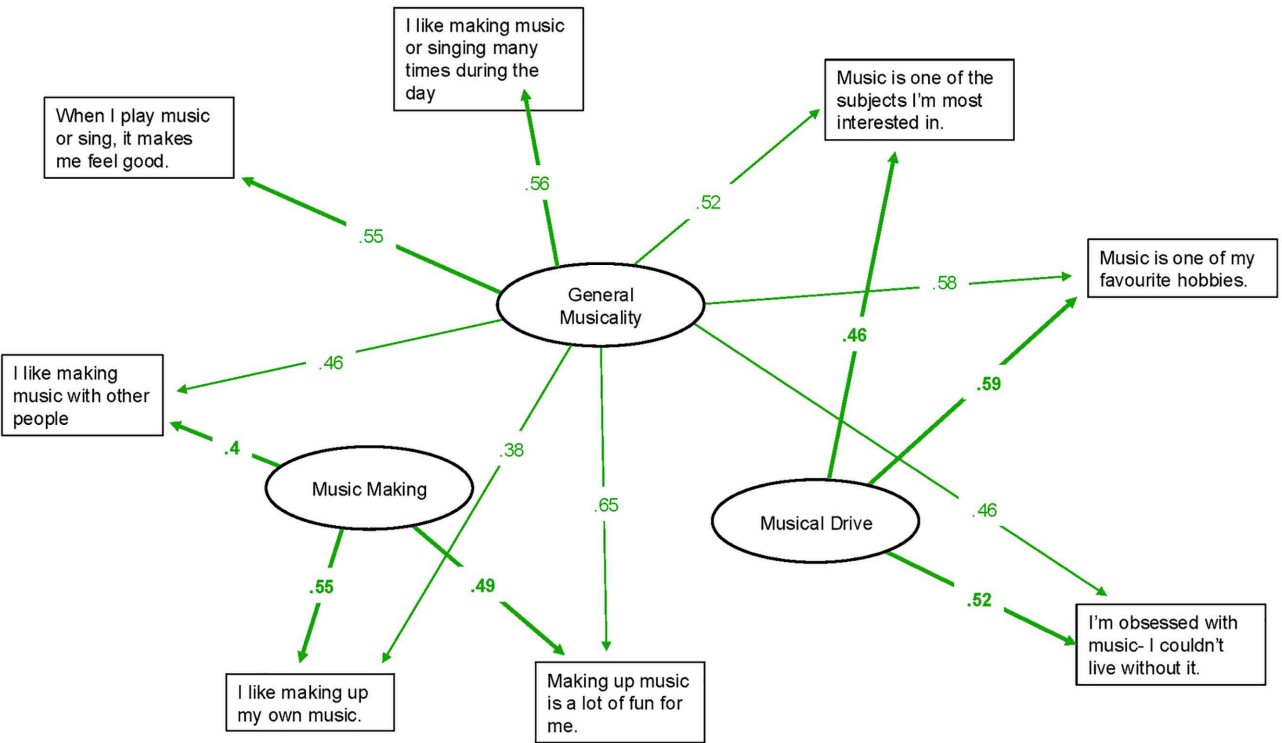

**Fig 1. Factor structure and items.** Factor loadings are indicated by the values of arrows.

**Table 1. Factor loadings for the 8 items included in the new scale.**

| Item number | Item statement | Musical Drive | Music Making | General Musicality |
|---|---|---|---|---|
| 1 | Music is one of my favourite hobbies. | **.59** | -<.001 | .58 |
| 2 | I'm obsessed with music – I couldn't live without it. | **.52** | −.04 | .46 |
| 3 | Music is one of the subjects that I'm most interested in. | **.46** | .08 | .52 |
| 4 | I like making up my own music. | −.16 | **.55** | .38 |
| 5 | Making music is a lot of fun for me. | .18 | **.49** | .65 |
| 6 | I like making music with other people. | .08 | **.4** | .46 |
| 7 | I like making music or singing many times during the day. | .23 | .36 | .56 |
| 8 | When I play music or sing, it makes me feel good. | .33 | .24 | .55 |

**Table 2. Reliability estimates for each of the factors.**

| Factor | Cronbach's alpha | MacDonald's omega |
|---|---|---|
| General Musicality (N = 225) | .84 | .89 |
| Musical Drive (N = 261) | .79 | .79 |
| Music Making (N = 244) | .71 | .73 |

the scale than their older counterparts, evidenced by a greater proportion of random and extreme responding in these age groups. Unexpectedly, we also found that 13-year-olds elicited more random and extreme responses than 8–12 year olds, on average. Nevertheless, we found that >50% of data from 6-, 7- and 13-year-olds did not exhibit patterns of random or extreme responding, providing preliminary evidence that responses from these age groups could be considered valid overall.

This analysis therefore indicated that the new 8-item scale was suitable for our target population (6–13 years), subject to validation against related measures.

## Study 1b

Study 1b aimed to investigate the external validity of the 8-item scale identified in study 1a by evaluating factor scores against parent and teacher ratings of child musicality and measures of children's music perception skills.

### Method

**Participants.** In total 250 children aged 6–13 took part in study 1b ($M = 9.2$, $SD = 2.02$, 126 female, 123 male, 1 non-binary). Data from children who were recruited as part of the US sample for study 1 were also used for this study (N = 185 aged 6–13, $M = 9.38$, $SD = 2.35$, 90 female, 94 male, 1 non-binary). New data were collected from 65 children from 3 primary schools in London aged 6–11 ($M = 8.68$, $SD = 1.35$; 36 female, 29 male). This wave of UK data collection took place between 30/01/2023 and 6/07/2023.

**Materials.** Research tools employed for the purposes of external validation differed by lab. In Las Vegas, children took part in: a musical emotion perception task [37], a task measuring sensitivity to musical harmony and tonality [38], two tasks measuring different aspects of musical beat perception [39,40] and a new musical reactions task which involved rating valence and arousal during music listening. Parents completed a version of the musical training subscale of the Gold-MSI (7 items) which reported on their child's level of musical training and were also asked to estimate the number of hours their children typically spent playing music for fun per day.

The Musical Emotions Perception Task (MEPT [37]) presented children with 30 short, affective auditory bursts expressing happiness, sadness, or fear, selected from two validated stimulus sets: the Montreal Affective Voices [41], consisting of nonverbal vocalizations, and the Musical Emotional Bursts [42] consisting of brief musical motifs played by a violin or clarinet. As in prior research using both stimulus sets in a behavioural task with adults [37], on each trial children heard one clip and were asked to select the intended emotion as either happy, sad, or fearful, with each emotion accompanied by a cartoon character displaying that emotion. Children were given practice trials with feedback before beginning the task, but they did not receive feedback on the main task.

The tonality perception task was framed as a puppet concert, in which children gave prizes to puppets based on who played the best song. Children watched eight videos in which each of two puppets played a melody or a chord sequence that ended on either the most expected note or chord according to rules of Western harmony (standard), on an out-of-key note or chord (unexpected key), or on a note or chord that was not expected but in the same key (unexpected harmony). On all trials one puppet played a standard sequence, and the other played a deviant sequence with either unexpected key or unexpected harmony. The tonality task was initially used to demonstrate sensitivity to Western tonality in 4- and 5-year-old children [38], a finding that was successfully replicated using the same stimuli in an electroencephalography (EEG) paradigm with even younger children [43].

For the Beat and Meter Sensitivity (BMS) task, children listened to 30 trials of expressively performed ballroom dance music and were asked to judge the fit of a metronome that could match the music at either the beat level, the measure level, both beat and measure levels, or neither level [40]. This task was initially used to document developmental changes in sensitivity to both beat and meter from age 5–17, results that were replicated with a separate sample of children using both the same task and stimuli as well as audiovisual versions of those stimuli [44].

While the BMS task measured how well children could detect correspondence between music and an overtly presented beat pattern (a metronome), the Internal Beat Perception Task (IBPT) assessed how well they could maintain a previously established beat pattern while listening to a beat-ambiguous stimulus. This type of beat perception holds relevance for hearing beat in syncopated rhythms and may contribute to the experience of musical groove [39]. In this task, children first hear a musical excerpt that robustly induces either a duple or a triple beat pattern, followed by an ambiguous melodic pattern that can be heard as conforming to either a triple or duple beat pattern. After up to 30 seconds of hearing the ambiguous pattern, children are asked to indicate the "correctness" of a probe drum beat that matches or mismatches the originally induced beat pattern (they are asked to judge how well a student drummer plays to the music). The IBPT task and stimuli was used to measure behavioural and EEG correlates of internal beat perception in adults [45] as well as with a separate sample of children aged 4–17 and adults [39].

For the musical reactions task, we adapted a recently developed emotional reactions task for use with children. Five video clips of live performed music were selected iteratively by a small group of young adult listeners to represent a range of classical, popular, film, and TV recordings that were likely to elicit musical chills. After each video, we presented the self-assessment manikin (SAM), developed and validated by Bradley & Lang [46], in which children were prompted to provide a rating of valence ("how would you rate the pleasantness of this video?") and arousal ("How relaxed or stimulated did you feel in response to the video") by selecting from five cartoon animations representing response options from "extremely unpleasant/relaxed" to "extremely pleasant/stimulated." This task thus yielded measures of mean valence and arousal of emotional reactions to music. The SAM has been used widely in research with children [47–49].

In London, data were collected using the Musical Sequence Transcription Task (MSTT [50]) and the Child Musicality Screening (CHIMUS [19]) which included both parent and teacher scales. For the cMI and MSTT, children's responses were collected on tablets. Children were required to read and respond to the items independently but had access to support from researchers when requested.

The MSTT [50] assesses children's ability to transcribe auditory sequences. Two guitar chords, one high and one low, are presented in different orders to create sequences of four. The task is to remember and write down which order the chords are presented in.

The MSTT was adapted for the current study. Sequences were presented on bongo drums, one high and one low, which were hidden from children's view in order to conceal visual cues. Four practice trials and 10 experimental trials were run in the current test sessions. Sequences were adjusted to be suitable for our age group on the basis of an Item Response Theory (IRT) analysis carried out by MSTT developers which provided estimates of item difficulty.

The Child Musicality Screening (CHIMUS [19]) is a 9-item scale developed for the purpose of rating children's musicality. Parents and teachers are asked to think of their child or a child from their class and rate agreement (strongly disagree- strongly agree) or frequency (never- always) in correspondence with statements such as 'The child has a great enthusiasm for music' on a Likert scale of 1–5. The 9 items are broken down into three subscales, each of which summarises children's perception skills, production skills or motivation for music.

**Procedure.** The research procedure differed by lab. In Las Vegas, responses to all tasks were collected on a computer. The order in which tasks were presented in was counterbalanced using a Latin Square design among the sample. For children ages 6–9, instructions and questions were read aloud by the experimenter as well as recorded. Children ages 10–13 were instructed to take each task on their own, with the experimenter monitoring to answer any questions.

In London, groups of 3–10 children at a time took part in a 30–45 minute session held in their music classroom or school library. The session began with the MSTT (15m), followed by a musical emotion listening task (10m) and the cMI (5-10m). All children read and responded to the cMI independently. Six-year-olds were tested in maximum groups of 6 to ensure that they could access additional help if needed. Following the tasks, children were given a sticker as a reward for their participation along with the parent CHIMUS to take home and return in a following research session. The teacher CHIMUS was also distributed following this session.

**Analysis.** Correlational analyses were carried out to evaluate the convergent validity of cMI factor scores. Effect sizes were interpreted as .1, .3 and .5 for small, moderate and large correlations [51].

The 8-items identified in study 1a target motivational and emotional components of music making. Given this, we expected to find large (*r* > .5) positive correlations between the cMI and informant reports regarding children's motivation for taking part in musical activities. We also expected to see a moderate (*r* > .3) relationship between cMI scores and performance on tasks assessing the ability to perceive and understand musical emotions.

On the basis of a theoretical relationship between motivation and ability (see: Self-Determination Theory, for example [52]), informant reports of child perception and production abilities were expected to be correlated to a small-moderate degree (*r* > .2) with scores on the 8-item cMI. We similarly expected to see small-moderate (*r* > .2) correlations between cMI scores and music perception tasks.

## Results

Correlations were computed between cMI scores and all additional measures. In some cases, questionnaires had not been received from parents and teachers and tasks had not been completed by children, thus resulting in varying sample sizes between pairs of measures. For correlation matrices which include Pearson's r values, sample sizes and significance reports see Table 3 for parent reports, Table 4 for teacher reports and Table 5 for music perception measures.

**Informant reports.** As expected, large correlations were found between children's self-reported general musicality/musical drive and informant reports of children's motivation for music (*r* = .52−.68). Self-reported enjoyment of music making was correlated to a lesser, moderate extent with informant-reported motivation for music (*r* = .33−.37). The general musicality and musical drive scores also exhibited a large overlap with teacher reports of perception and production skills (*r* = .4−.63). They were only correlated to a small-moderate extent with parent reports of perception and production skills and these correlations did not reach significance (*r* = .21−.31).

Moderate, significant correlations were uncovered between all cMI factor scores and parent reports on the number of hours children spent playing music for fun per day (*r* = .21−.25). Parent reports on children's level of musical training exhibited a small yet significant correlation with children's cMI scores (*r* = .18−.19).

**Table 3. Correlations between cMI scores and PARENT reports on children's musicality.**

| | Musical Drive (cMI) | Music Making (cMI) | General Musicality (cMI) | Musical training (GoldMSI) | Recreation hours | Motivation (CHIMUS) | Perception (CHIMUS) | Production (CHIMUS) |
|---|---|---|---|---|---|---|---|---|
| **Musical training (GoldMSI)** | .18* (185) | .18* (185) | .19* (185) | 1 | – | – | – | – |
| **Recreation hours** | .22* (92) | .21* (92) | .25* (92) | .38 (92) | 1 | – | – | – |
| **Motivation (CHIMUS)** | .53* (47) | .33* (47) | .52* (47) | NA | NA | 1 | – | – |
| **Perception (CHIMUS)** | .27 (49) | .25 (49) | .28 (49) | NA | NA | .39* (47) | 1 | – |
| **Production (CHIMUS)** | .11 (46) | .14 (46) | .16 (46) | NA | NA | .26 (45) | .26 (46) | 1 |

Note. Sample size (N) is displayed in brackets following *r* values. * indicates significance at an alpha level of 0.05.

**Table 4. Correlations between cMI scores and TEACHER reports from the Child Musicality Screening (CHIMUS).**

| | Musical Drive (cMI) | Music Making (cMI) | General Musicality (cMI) | Motivation (CHIMUS) | Perception (CHIMUS) | Production (CHIMUS) |
|---|---|---|---|---|---|---|
| **Motivation (CHIMUS)** | .68* (33) | .37* (33) | .61* (33) | 1 | – | – |
| **Perception (CHIMUS)** | .63* (34) | .31 (34) | .56* (34) | .96* (33) | 1 | – |
| **Production (CHIMUS)** | .47* (34) | .21 (34) | .4* (34) | .84* (33) | .89* (34) | 1 |

Note. Sample size (N) is displayed in brackets following *r* values. * indicates significance at an alpha level of 0.05.

**Table 5. Correlations between cMI scores and performance on music perception tasks.**

| | Musical Drive (cMI) | Music Making (cMI) | General Musicality (cMI) | Beat & Meter Sensitivity (BMS) | Internal Beat Perception Task (IBPT) | Tonality perception | Musical Emotion Perception Task (MEPT) | Musical reactions (valence) | Musical reactions (arousal) | Musical sequence transcription task (MSTT) |
|---|---|---|---|---|---|---|---|---|---|---|
| **Musical Drive** | 1 | – | – | – | – | – | – | – | – | – |
| **Music Making** | .47*(250) | 1 | – | – | – | – | – | – | – | – |
| **General Musicality** | .86* (250) | .82* (250) | 1 | – | – | – | – | – | – | – |
| **BMS** | .22* (185) | <.01 (185) | .13 (185) | 1 | – | – | – | – | – | – |
| **IBPT** | .06 (183) | .09 (183) | .1 (183) | .18* (183) | 1 | – | – | – | – | – |
| **Tonality perception** | .05 (185) | −.01(182) | .03 (182) | .28* (182) | .16* (183) | 1 | – | – | – | – |
| **MEPT** | .17* (183) | .04 (183) | .12 (183) | .35* (183) | .12 (181) | .19* (183) | 1 | – | – | – |
| **Valence reactions** | .26* (180) | .16* (180) | .25* (180) | .16* (180) | .05 (179) | .09 (180) | .18* (180) | 1 | | – |
| **Arousal reactions** | −.05 (180) | .07 (180) | <.01 (180) | −.12 (180) | .07 (179) | −.11 (180) | −.17* (178) | −.3* (180) | 1 | – |
| **MSTT** | .24 (64) | .11 (64) | .19 (64) | NA | NA | NA | NA | NA | NA | 1 |

Note. Sample size (N) is displayed in brackets following *r* values. * indicates significance at an alpha level of 0.05.

**Music perception tasks.** Children's ratings of positive valence in a music listening task were correlated to a small to moderate extent with their self-reported musicality, musical drive and enjoyment of music making (*r*=.16−.26). Self-reported musical drive was also correlated to a small to moderate extent with performance in the beat perception task (BMS; *r*=.22) and the musical emotion perception task (*r*=.17).

No significant correlations were found between cMI scores and ability to transcribe musical sequences (MSTT), arousal ratings in the musical reactions task or performance in the tonality or internal beat perception (IBPT) tasks.

## Study 1c

Study 1b provided evidence for the validity of the cMI. The aim of study 1c was to derive data norms from a larger sample and produce better informed estimates of task reliability.

### Method

Data from studies 1a-c (N=347) were supplemented with data gathered online from a new sample of children in the UK between 15/06/24 and 08/01/2025. The new sample (N=113) followed the same procedure as outlined in the study 1 methods section for children based in London. They were 6–11 years of age (*M*=8.21, *SD*=1.65; 65 female, 46 male, 1 'other' and 3 who did not provide gender information). Combined with data from previous studies, the final sample consisted of 460 children aged 6–13 (*M*=8.81, *SD*=2.01; 236 female, 217 male, 1 'other', and 6 who did not provide gender information).

### Analysis

Reliability estimates were calculated for the full scale and the two sub-scales. We included Cronbach's 'alpha', a frequently reported estimate of internal consistency, along with two other measures: McDonald's 'omega' [53] and Guttman's lambda 6 [54], a lower-bound estimate. Estimates were calculated using the *alpha* and *omega* functions from the psych package in R [35].

### Results

Reliability analyses provided evidence for the 'good' reliability of the general musicality scale (α=.84, ω=.89) and the musical drive (α=.79, ω=.8) subscale. The reliability of the music making subscale was 'acceptable' (α=.68, ω=.7). Table 6 provides a breakdown of descriptive statistics and reliability estimates in age brackets.

**Table 6. Summary statistics and reliability estimates for the new child Musicality Index (N=460).**

| Age (N) | General Musicality | | | | Musical Drive | | | | Music Making | | | |
|---|---|---|---|---|---|---|---|---|---|---|---|---|
| | 6-7 (143) | 8-9 (145) | 10-11 (123) | 12-13 (49) | 6-7 (142) | 8-9 (141) | 10-11 (109) | 12-13 (49) | 6-7 (121) | 8-9 (129) | 10-11 (109) | 12-13 (49) |
| Mean (SD) | 3.51 (.87) | 3.57 (.81) | 3.64 (.8) | 3.53 (.84) | 3.21 (1.13) | 3.38 (1.02) | 3.5 (.97) | 3.61 (1.12) | 3.71 (.92) | 3.58 (.86) | 3.65 (.87) | 3.21 (1) |
| alpha | .83 | .86 | .83 | .87 | .74 | .78 | .84 | .86 | .59 | .67 | .7 | .79 |
| omega.tot | .87 | .9 | .89 | .92 | .75 | .79 | .86 | .87 | .62 | .73 | .72 | .83 |
| G6 | .83 | .86 | .86 | .9 | .66 | .71 | .8 | .81 | .51 | .6 | .62 | .75 |

All subscales had a minimum of 1 and a maximum of 5, as scores were calculated by taking an average of responses on a 5-point Likert scale.

Study 1c established the reliability of the scale across the age group of interest in a large sample. The reliability of the music making scale was below commonly accepted levels ($\alpha < .7$) in the youngest age bracket (6 & 7 years).

## Study 2

Studies 1a-1c established 8 items for a new, short assessment of child musicality, and provided estimates of the psychometric properties of this short scale. Study 2 recruited a new sample to further investigate test validity and reliability. This study also aimed to assess whether the scale could provide valid and reliable psychometric estimates for the youngest target population (6- and 7-year-olds).

### Method

Data were collected from a new sample of children in the US between 25/7/2023 and 16/2/2025. As opposed to the samples in studies 1a-1c who saw a larger selection of items, the new sample (N=56) responded to the 8-item cMI. Children additionally answered two further questions targeting other musical behaviours, instrument playing and perfect pitch, which were not part of the 8-item cMI. They completed the scale alongside several perceptual measures which were administered as part of a separate study. Children aged 6–9 (N=38) had instructions read aloud and their answers recorded by an assisting adult, children aged 10–11 (N=18) read and responded to items independently. They were 6–11 years of age ($M=8.67$, $SD=1.88$; 26 female, 29 male, 1 'other').

### Analysis

Reliability was assessed using the same three indicators outlined in study 1c. Response bias was investigated by testing for patterns of extreme and random responding, employing the same methodology as in study 1a.

### Results

Study 2 analysed responses from a sample who completed the new 8-item scale (N=56). Reliability estimates were 'good' for both the general musicality scale ($\alpha=.84$; $\omega=.9$) and the musical drive scale ($\alpha=.85$; $\omega=.86$), and 'acceptable' ($\alpha=.69$; $\omega=.73$) for the music making subscale (see Table 7).

We lastly investigated whether response bias differed by age group. Visual inspection of average proportion of random and extreme responses by age group indicated that 6- and 7-year-olds demonstrated similar levels of random and extreme responding as children in older age groups. Taking response bias as an indicator of understanding, this finding indicated that 6- and 7-year-olds can understand the scale as well as children 8 years and above. A complete outline of this analysis is provided in the S4 File.

In sum, study 2 provided further evidence for the psychometric properties of the cMI across the full age range of its target population (6–13 years).

 

**Table 7. Reliability estimates for each of the factors.**

| Factor | Cronbach's alpha | MacDonald's omega |
|---|---|---|
| General Musicality (N = 56) | .84 | .9 |
| Musical Drive (N = 56) | .86 | .87 |
| Music Making (N = 56) | .69 | .73 |

## Discussion

The assessment of music during childhood has too often been reduced to simple auditory perception tasks which do not adequately capture the complexities of childhood musicality. Parent and teacher questionnaires can provide a more nuanced view of childhood musicality, but at the same time may present issues with informant bias. Children's self-reports present a valuable opportunity to validate informant reports and add to information gathered using ability measures. The new 8-item cMI presents researchers with a psychometric scale for assessing aspects of musicality that cannot be captured by perceptual ability tasks and which are especially relevant during childhood.

**Facets of child musicality.** Study 1a successfully identified a subset of 8 items to include in the new self-report scale. These items contributed to a simple and well-fitting factor solution with a general factor and two group factors, each consisting of 3 items. The new 8-item child Musicality Index (cMI) thus provides an overall 'General Musicality' estimate and two sub-scores estimating children's 'Musical Drive' and enjoyment of 'Music Making'.

In contrast to the five facets of musical sophistication in adulthood assessed by the Gold-MSI, the current investigation highlighted these two key areas of musical drive and enjoyment of music making as particularly relevant during childhood. While previous theory has noted the importance of these facets [55,56], there has, until now, been a lack of self-report tools appropriate for capturing and studying them with children who have received no training on a musical instrument. Given that instrumental learning is not always available to children, the current assessment of motivation for and enjoyment of music making provides a more inclusive tool that is appropriate for all children regardless of training and is therefore better suited to capture musicality as it manifests during childhood.

In line with previous research that indicates singing, dancing and musical play are better indicators of interest in music than instrument playing [57], items from the original GoldMSI which related to musical training did not contribute to the final factor solution whereas items adapted from the active engagement (AE) Gold-MSI scale did. Items from the AE scale were retained as part of the 'musical drive' factor and focused on music as a favourite subject, a favourite hobby or an 'obsession'. This factor therefore touches on identification with music and perception of self as well as relating to motivation for engaging with music. An additional item adapted from the AE scale 'I like making music or singing many times during the day' was included in the general factor, which alludes both to motivational aspects of musicality, captured by the 'musical drive' factor in our model, as well as relating to the enjoyment of music making, captured by the 'music making' factor.

The 3 items in the music making scale originated from the MCQ. They speak to the enjoyment of creating music alone or with others. Interestingly, the Gold-MSI does not ask about composing music, focusing more on music listening as well as the analysis of and quality of engagement with music. This leads us to speculate that the enjoyment of music making dictates musicality in childhood to a much greater extent than it does musical sophistication in adolescence and adulthood, with music listening becoming more relevant as we get older.

The pattern of prevalence of music making over music listening in childhood revealed here was unwittingly reflected in the development of the final item included in the scale. The item: 'When I play music or sing, it makes me feel good', included in the general factor was adapted from the emotional engagement scale of the Gold-MSI. The original item 'I often pick certain music to motivate or excite me' could relate to either choosing music to play or listen to, while the adapted version clearly designates music making. The adapted item also shifts focus to the enjoyment of a musical

experience, which is in line with previous literature suggesting that experience of musical emotions plays a key role in motivating music-making during childhood [58].

**Internal validity.** Response biases have been shown to influence self-reports gathered from younger children (5–6 years) [29,59–61]. Since it is possible that such biases could impact the validity of the cMI, we checked on the degree of random and extreme responding in studies 1 and 2. Study 1a indicated that the youngest participants in our sample (6- & 7-year-olds) were the most likely to exhibit random and extreme responses. In order to ascertain whether random and extreme responses to the cMI identified in study 1a would persist with the shorter 8-item scale, we carried out study 2 with a new sample. In this group, all of the children between 6–9 years of age received one-to-one supervision. This second study demonstrated that 6- and 7-year-olds responded with a similar level of validity to children in the older groups (8–11). These results suggest that response biases identified in studies 1a-c could reasonably be explained by the fact children saw a large set of items (up to 48 at a time) and were more likely to provide random/inconsistent or extreme responses as a consequence of fatigue. We nevertheless recommend that 6- and 7-year-olds complete the 8-item cMI with one-to-one supervision to reduce the chances of random and extreme responding with in these younger age groups [21].

Our investigation into response patterns in study 1a also revealed a U-shaped trend, whereby participants of 13 years generated nearly the same proportion of random and extreme responses as 6-year-olds. Whilst unexpected, we speculate that 13-year-olds could have found the questions too repetitive or perceive the scale to be aimed at children younger than them, preventing them from always taking their answers seriously. As a result, it is recommended that researchers consider using the Gold-MSI alongside the cMI with this age group to ensure they are picking up on all relevant facets of musicality and musical sophistication during the transition from childhood to adolescence.

**External validity.** To assess the external validity of the new measure, study 1b investigated the relationship between cMI scores, informant reports and music perception tasks. Findings broadly adhered to our hypotheses and provided some preliminary evidence for the convergent validity of the cMI. All factors were correlated with informant reports regarding children's motivation for taking part in musical activities and thus provided some strong evidence for external validity with expected moderate to large effect sizes. Additional evidence for the validity of the musical drive and general musicality factors was provided by large correlations with teacher ratings of children's perception and production skills. Correlations with parent reports of perception, production, children's number of hours spent playing music for fun and musical training were small-moderate, in comparison.

The distinction between correlations of the cMI with teacher and parent reports could suggest that teachers can more easily pick up on children's motivation for music and better evaluate children's perception and production skills compared with parents. This could be explained by the fact that teachers regularly interact with many children and are therefore likely to have a better frame of reference for making comparative judgements. Teachers may also have more opportunities than parents to observe children engaged in musical activities given the compulsory nature of music education in many school settings. Future research should endeavor to collect teacher reports in addition to parent and child reports to ensure the best chance of gathering an accurate picture of child musicality.

The cMI was also significantly correlated with children's performance on tasks involving perceiving and experiencing emotions during music listening. Notably, children's self-reported musical drive was related to their performance on an emotion perception task and their ratings of positive valence during the musical reactions task. These ratings of positive valence were also significantly associated with scores on the music making and general musicality factors, suggesting children with higher scores experienced more enjoyment while listening to music. This pattern substantiates the idea that motivation for music could be affected by the ability to perceive, enjoy and play it [58].

In contrast, there were no significant associations found between cMI factors and tonality, internal beat perception, arousal ratings in the music reactions task and the ability to transcribe rhythmical sequences. This finding supports the notion that perceptual abilities do not always account for the variability in children's enjoyment of and drive to engage with

music, and affirms that the cMI is picking up on something additional to the tasks which are frequently employed in developmental research into musicality [e.g., 15].

It should be highlighted that the small and varied sample sizes in the current study make findings difficult to interpret. Further exploration with larger samples and varied measures of musicality should be carried out to better assess the convergent validity of the new cMI. This will be necessary to fully validate the new instrument.

**Reliability.** Studies 1c and 2 indicated 'good' ($\alpha > .8$, $\omega > .7$) reliability of the general musicality and musical drive subscales and 'acceptable' reliability of the music making factor ($\alpha = .68 - .69$, $\omega > .7$). One issue which should be noted, however, is the 'questionable' to 'unacceptable' ($\alpha = .59$, $\omega = .62$) reliability of the music making subscale with groups of younger children (6 and 7 years; see Table 6). Given that children in the study 1c sample saw at least 24 items at a time, it is hoped that results from a larger future sample with the short 8-item cMI will improve upon the current estimates. Future studies should therefore confirm whether this subscale is psychometrically robust across the full age range. Meanwhile, we suggest that researchers check on subscale reliability with their own data and interpret the music making subscale with caution when assessing 6- and 7-year olds.

## Recommendations for research use

Based on the findings of studies 1 and 2 we recommend that researchers administering the cMI to 6- and 7-year-olds (1) provide one-to-one supervision and (2) collect informant reports in addition to self-reports. For 13-year-olds we recommend administering the task alongside the original Gold-MSI to gather an appropriate evaluation of musicality in the transition from childhood to adolescence. It would be useful to investigate this use of the two scales in tandem in future studies to determine which measure is likely to capture the most meaningful variation in musical behaviour in this age group.

The data reported in the current investigation were collected from a heterogeneous sample of participants from two different English-speaking countries with different socio-cultural backgrounds. This heterogeneity should strengthen the generalisability of the instrument to other English-speaking samples and populations.

Moreover the cMI should be suitable for a wide range of children since it includes simple language and relatively few items. Nonetheless, researchers are likely to benefit from considering how they present the cMI to make it appropriate for their sample. Options to improve accessibility could include reading out items out loud or making use of visually represented response options.

Data were not collected on the developmental profiles of children in the current investigation, and we were consequently unable to investigate how well the new instrument captures musicality with neurologically diverse groups of children. Future studies should endeavour to gather data norms and reliability estimates for populations with atypical developmental trajectories.

## Future directions

The new cMI could contribute to a more inclusive and detailed understanding of child musicality than has been achieved by questionnaires on instrumental learning or perceptual measures in the past. Data from the cMI can be used to explore the interactions between musicality and other important aspects of child development, such as language and cognitive skills. Furthermore, by facilitating the accumulation of evidence on musicality through early and late childhood, we hope that insights from the cMI can be used to strengthen the argument for access to music across these critical developmental stages.

It is important to note that to establish a comprehensive understanding of the rich, varied nature of musicality, self-report and informant-report scales must be complemented with ability tasks that can capture musical activities as they naturally manifest during childhood. To further facilitate future progress in this area we therefore encourage the development of novel and creative ability measures that can assess children's musical drive and enjoyment of music making while maintaining the high psychometric standards required for individual differences research.

## Conclusion

The new child Musicality Index (cMI) is a short, 8-item, psychometric self-report tool for children aged 6–13 that has been adapted from the adult Goldsmiths Musical Sophistication Index. The scale has been designed to assess social, emotional and motivational components which have been considered critical to musicality during childhood, but have rarely been accounted for. It is suitable for all children, regardless of whether they have received instrumental training, and freely available for research use. By applying the cMI in future studies, researchers will be able to gain further insight into interactions between musicality and other important aspects of child development.

## Supporting information

**S1 Data. Edited and original statements from study 1a.**
(CSV)

**S2 Data. Response formats for edited and original Items from study 1a.**
(CSV)

**S3 File. Response validity analysis from study 1a.**
(DOCX)

**S4 File. Response validity analysis from study 2.**
(DOCX)

**S5 File. Inclusivity in global research questionnaire.**
(DOCX)

## Acknowledgments

We would like to thank Sivan Barashy, Jennifer Tipple, Lucy-Ellen Parker, Theresa Freeburn, Sarah Soyler, and Simon Lambert for assisting with and facilitating data collection.

## Author contributions

**Conceptualization:** Chloe MacGregor, Solena Mednicoff, Erin Hannon, Daniel Müllensiefen.

**Data curation:** Chloe MacGregor, Solena Mednicoff, David J. Vollweiler, Erin Hannon.

**Formal analysis:** Chloe MacGregor, Daniel Müllensiefen.

**Investigation:** Chloe MacGregor, Solena Mednicoff, David J. Vollweiler.

**Methodology:** Chloe MacGregor, Solena Mednicoff, Erin Hannon, Daniel Müllensiefen.

**Project administration:** Erin Hannon.

**Resources:** Chloe MacGregor, Solena Mednicoff, Erin Hannon, Daniel Müllensiefen.

**Supervision:** Erin Hannon.

**Validation:** Chloe MacGregor.

**Visualization:** Chloe MacGregor.

**Writing – original draft:** Chloe MacGregor.

**Writing – review & editing:** Chloe MacGregor, Solena Mednicoff, David J. Vollweiler, Erin Hannon, Daniel Müllensiefen.

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
