## [Decision Letter · Decision Letter 0]

6 Jun 2025

Dear Dr. MacGregor,

Thank you for submitting your manuscript to PLOS ONE. After careful consideration, we feel that it has merit but does not fully meet PLOS ONE’s publication criteria as it currently stands. Therefore, we invite you to submit a revised version of the manuscript that addresses the points raised during the review process.

**There would likely be much interest in a children's validated version of the Goldsmith's MSI, and the research reported on here is timely and relevant. However, both reviewers point out significant limitations in the current study and its reporting, suggesting the need for further clarity in the analyses. The most significant concern is a lack of validation of the final instrument, which does not square with the conclusions in the abstract/discussion that this is a useful tool. Validation of the proposed short tool in a new sample would be useful. The reviewers also noted patterns in the results that warrant acknowledgement/discussion, as well as concerns that the scale is implemented across a wide developmental age range without adjustment.**

We look forward to receiving your revised manuscript.

Kind regards,

Jessica Adrienne Grahn

Academic Editor

PLOS ONE

**Journal Requirements:**

1. When submitting your revision, we need you to address these additional requirements. Please ensure that your manuscript meets PLOS ONE's style requirements, including those for file naming. The PLOS ONE style templates can be found at https://journals.plos.org/plosone/s/file?id=wjVg/PLOSOne_formatting_sample_main_body.pdf and https://journals.plos.org/plosone/s/file?id=ba62/PLOSOne_formatting_sample_title_authors_affiliations.pdf 2. Please include a complete copy of PLOS’ questionnaire on inclusivity in global research in your revised manuscript. Our policy for research in this area aims to improve transparency in the reporting of research performed outside of researchers’ own country or community. The policy applies to researchers who have travelled to a different country to conduct research, research with Indigenous populations or their lands, and research on cultural artefacts. The questionnaire can also be requested at the journal’s discretion for any other submissions, even if these conditions are not met.  Please find more information on the policy and a link to download a blank copy of the questionnaire here: https://journals.plos.org/plosone/s/best-practices-in-research-reporting. Please upload a completed version of your questionnaire as Supporting Information when you resubmit your manuscript. 3. When completing the data availability statement of the submission form, you indicated that you will make your data available on acceptance. We strongly recommend all authors decide on a data sharing plan before acceptance, as the process can be lengthy and hold up publication timelines. Please note that, though access restrictions are acceptable now, your entire data will need to be made freely accessible if your manuscript is accepted for publication. This policy applies to all data except where public deposition would breach compliance with the protocol approved by your research ethics board. If you are unable to adhere to our open data policy, please kindly revise your statement to explain your reasoning and we will seek the editor's input on an exemption. Please be assured that, once you have provided your new statement, the assessment of your exemption will not hold up the peer review process. 4. Please amend either the abstract on the online submission form (via Edit Submission) or the abstract in the manuscript so that they are identical.

Reviewers' comments:

Reviewer's Responses to Questions

**Comments to the Author**

1. Is the manuscript technically sound, and do the data support the conclusions?

Reviewer #1: Yes

Reviewer #2: No

2. Has the statistical analysis been performed appropriately and rigorously?

Reviewer #1: Yes

Reviewer #2: I Don't Know

3. Have the authors made all data underlying the findings in their manuscript fully available?

Reviewer #1: Yes

Reviewer #2: Yes

4. Is the manuscript presented in an intelligible fashion and written in standard English?

Reviewer #1: Yes

Reviewer #2: No

**Reviewer #1:**  In the present study, the authors develop a child-directed version of the Goldsmiths Musical Sophistication Index—a self-report questionnaire widely used to assess individuals' degree of musical sophistication, with valuable contributions to the field of cognitive science. I believe that adapting this tool for use in developmental populations is a valuable and timely contribution. This is particularly relevant given the growing body of research highlighting the important role of music in supporting the development of various cognitive skills, especially those related to language. I have only a few minor comments for the authors, which are detailed below.

1. I found the abstract somewhat difficult to follow, particularly the sections describing the factor analysis and reliability. The language is quite technical and may not be easily understood by readers who are not familiar with these analytical approaches. I would recommend reserving the technical details for the Methods section and focusing on delivering a clear and accessible summary of the main findings and implications in the abstract.

2. The choice of the broad age range (6–13 years) needs to be better justified. A great deal is being asked of the test to function equally well across both early and late childhood, which represent developmentally distinct populations. To name just a few developmental differences, some critical white matter pathways, such as the arcuate fasciculus, do not fully mature until around age 8, and children transition from non-readers to proficient readers during this period. These substantial changes suggest that it may be more appropriate to develop two separate versions of the test—one for early childhood (e.g., 4–8 years) and another for late childhood (e.g., 9–13 years). This could also help explain some of the challenges the authors report in their results, such as lower reliability at the age extremes (6 and 13 years). As they themselves note, some items may seem too childish for 13-year-olds, while the use of a Likert scale may not be appropriate for 6-year-olds

3. Regarding readability, the ATOS test assigns a reading age of 8–9 years, which suggests that children aged 6 to 7 may struggle to fully understand the questionnaire items.

4. It is not clear whether younger children were expected to read and respond to the items independently, or whether the items were read aloud to them and their answers recorded by an assisting adult.

5. The authors describe the analyses within the Results section, which presents several drawbacks. On the one hand, it disrupts the flow of the results; on the other, it prevents a thorough and appropriately structured explanation of the analytical approaches and terminology. For example, I would appreciate a clearer explanation of the exploratory factor analysis, a more detailed description of the hierarchical omega and how it is computed, same for the guttman error/indice and clarification of the meaning of the "2+1 model." I therefore suggest moving the description of the analyses to the Methods section and including more detail to enhance transparency and reproducibility.

6. The final 8 items were selected from an initial pool of 45 items completed by the children. As the authors note, the results may differ in future assessments where only the 8-item version is administered. Therefore, I would like to see a comparison between the outcomes obtained using the full version and those from the final, shortened version. Did any participants in Study 2b complete the short version? If so, please compare their results with those who completed the full version. If not, it may be worthwhile to collect a small dataset—perhaps restricted to a specific grade level—and compare it to the results from the larger cohort that completed the full version. Doing so would substantially strengthen the validity of the proposed tool.

7. In Study 2b, the thresholds assigned to the correlations appear somewhat arbitrary. I recommend including the correlation analyses and clearly defining the expected outcomes in the Methods section to ensure transparency and reproducibility.

8. According to the initial analyses, the authors explicitly acknowledge that the assessment may not be valid for the extreme age groups (6 and 13 years old). However, in Study 2b, they still collected data across the entire age range, and in the general discussion, they state that the test is suitable for ages 6 to 13. This appears somewhat inconsistent and would benefit from further clarification.

9. line 704 page 32 “versus”?

10. Please include a description of the reliability assessment, as well as the alpha and omega parameters computed, in the Methods section for greater clarity and transparency.

**Reviewer #2:**  The ms has an unfinished feel to it, and unless I'm missing something, we don't have a test of the actual proposed 8-item scale that includes only children who took the final scale. Children from Study 1 seem to be included in all of the other studies. The General Discussion is also particularly revealing--2 pages of a 38-page manuscript that doesn't convince me that I should ever use the scale, or that the authors even think that it's particularly useful. The results also go on and on and on, making the ms a chore to read.

As for the rationale, it's probably true that objective tests miss out on some aspects of musicality in childhood, but the authors assert this without actually convincing the reader why objective tests are a problem.

I note that in Table 1 and Figure 1, 2 of 3 items in the Music Making subscale have higher correlations with the General Musicality score than they do with the actual subscale. So why is Music Making a subscale? Similarly, for Musical Drive, 1 of 3 items is more strongly correlated with General Musicality than it is with Musical Drive, and another item has virtually identical correlations with the aggregate and subscale scores. In other words, the model of musicality embodied by the scale doesn't seem to work. Or at least the hierarchy of levels is not consistent with Carroll.

The result that the data for the oldest group of children were as noisy as they were for the youngest children is disconcerting.

What is the quadratic regression equation? Why the mystery about using an unspecified "quadratic formula"?

With only 8 it

ems and marked individual differences in musicality, is it really extreme if a particularly musical child only chose responses of 1 or 5?

Several typos: just some examples.

Data are plural

l. 85: that accompanies PLAYING or LEARNING music as part...

It's used for its (possessive) in the first sentence of the ms. One of the authors is a native speaker of English...

I was excited to review this manuscript, and I like the Gold-MSI a lot, so I started the review with a positive bias.

**Do you want your identity to be public for this peer review?** For information about this choice, including consent withdrawal, please see our Privacy Policy

Reviewer #1: No

Reviewer #2: No

---

## [Author Response · Author response to Decision Letter 1]

23 Oct 2025

We would like to thank you for your thoughtful and constructive suggestions which have greatly improved the manuscript. We believe the revised version has benefitted in particular from justifying our choice of age range, clarifying how we carried out our analyses and adding a new study, which provides evidence for the psychometric properties of the final task in a new sample. You will also find that the discussion section has been expanded upon and the results sections are more concise, in line with your recommendations.

Please find detailed responses to each of your comments below. We hope that you enjoy reading the revised manuscript and look forward to your response.

Editors’ comments:

There would likely be much interest in a children's validated version of the Goldsmith's MSI, and the research reported on here is timely and relevant. However, both reviewers point out significant limitations in the current study and its reporting, suggesting the need for further clarity in the analyses. The most significant concern is a lack of validation of the final instrument, which does not square with the conclusions in the abstract/discussion that this is a useful tool. Validation of the proposed short tool in a new sample would be useful. The reviewers also noted patterns in the results that warrant acknowledgement/discussion, as well as concerns that the scale is implemented across a wide developmental age range without adjustment.

To address the concerns raised here we have (1) added further detail to clarify our analysis procedures, (2) expanded upon the discussion section to address patterns in the results which hadn’t been fully addressed and (3) included a new study which validates the final instrument in a new sample. Insights from the new study provide stronger evidence for the psychometric properties of the scale which will make it an even more attractive tool for researchers.

Reviewer 1 comments:

In the present study, the authors develop a child-directed version of the Goldsmiths Musical Sophistication Index—a self-report questionnaire widely used to assess individuals' degree of musical sophistication, with valuable contributions to the field of cognitive science. I believe that adapting this tool for use in developmental populations is a valuable and timely contribution. This is particularly relevant given the growing body of research highlighting the important role of music in supporting the development of various cognitive skills, especially those related to language. I have only a few minor comments for the authors, which are detailed below.

1. I found the abstract somewhat difficult to follow, particularly the sections describing the factor analysis and reliability. The language is quite technical and may not be easily understood by readers who are not familiar with these analytical approaches. I would recommend reserving the technical details for the Methods section and focusing on delivering a clear and accessible summary of the main findings and implications in the abstract.

We agreed that the abstract was too technical. The new edit (p. 2 lines 27-43) should provide a more accessible summary for readers who are unfamiliar with factor analysis, including additional detail on research implications as suggested.

2. The choice of the broad age range (6–13 years) needs to be better justified. A great deal is being asked of the test to function equally well across both early and late childhood, which represent developmentally distinct populations. To name just a few developmental differences, some critical white matter pathways, such as the arcuate fasciculus, do not fully mature until around age 8, and children transition from non-readers to proficient readers during this period. These substantial changes suggest that it may be more appropriate to develop two separate versions of the test—one for early childhood (e.g., 4–8 years) and another for late childhood (e.g., 9–13 years). This could also help explain some of the challenges the authors report in their results, such as lower reliability at the age extremes (6 and 13 years). As they themselves note, some items may seem too childish for 13-year-olds, while the use of a Likert scale may not be appropriate for 6-year-olds

The purpose of the current research is to provide a standardised quantitative assessment of musicality that could be used for comparing children across childhood, and we therefore aimed to include as wide an age range as possible. Our justification for including 6-year-olds this approach lies in similar, well-established scales which are used to assess musical behaviour in this age range, for example the motivation for learning music questionnaire (Comeau et al., 2019) which caters for ages 6-17.

On the use of Likert scales, we are aware of the greater difficulty that may be faced by 6-year-olds completing these scales in comparison to older children. However, in line with research into their use with this age group, we shortened the scales to 5-points to facilitate ease of responding. We considered using a 3-point scale, but decided to opt for 5 in accordance with research indicating that 5-6 year old children respond in a similar manner to 3- and 5-point Likert scales (Chambers & Johnston, 2002).

Furthermore, while research has indicated that responses on 5-point agreement and frequency scales may not perfectly reflect yes/no dichotomous response formats (Mellor & Moore, 2014), we are interested in investigating patterns of individual variation and preferred to prioritise an approach that could allow for greater discrimination between children’s responses.

In order to address your concerns, we have added justification for the decision to include a wide age-range to the outline of aims at the end of the introduction (p. 8 lines 294-296) and for our choice of Likert scales in the materials section of study 1a (p. 10 lines 233-239).

3. Regarding readability, the ATOS test assigns a reading age of 8–9 years, which suggests that children aged 6 to 7 may struggle to fully understand the questionnaire items.

Comprehension was a key concern for us due to the developmental differences you outlined above. For this reason, we made sure that the initial set of 48-items were simplified as much as possible. Given that our aim was to develop a children’s version of the Gold-MSI, however, we also wanted to ensure adapted items were still comparable to the original Gold-MSI items. This meant that we compromised with the average reading age estimate, which was higher than our intended sample age.

The final set of items have an estimate of 7-8 years average reading age, which means 6- and 7-year-olds may still struggle to understand the scale as you have highlighted. To test this expectation, we included all children in our study 1a response validity analyses to see whether children in these younger age groups responded differently to the scale. We indeed found that 6- and 7-year-olds displayed higher random/inconsistent and ‘extreme’ responding in some cases and as a result recommend additional support from adults for this age group to improve response validity.

Study 2 provides evidence for improved validity in new data from 6-7 year olds who completed the 8-item version of the scale with one-to-one supervision. We now emphasise these recommendations in the Discussion (p. 34 lines 680-683; p.36 lines 742-744) as well as making it clear that researchers should interpret cMI results with caution in these younger age groups.

4. It is not clear whether younger children were expected to read and respond to the items independently, or whether the items were read aloud to them and their answers recorded by an assisting adult.

In the US sample, children aged 6-9 had instructions read aloud and had their answers recorded by an assisting adult. Children aged 10-13 in the US sample and aged 6-13 in the UK sample read and responded to the items independently.

Further clarification of the procedure has been included in the methods section of study 1a (p. 11 lines 251-254 and 269-270), study 1c (p. 22 lines 492-494 and 497-499) and study 2 (p. 30 lines 596-599).

5. The authors describe the analyses within the Results section, which presents several drawbacks. On the one hand, it disrupts the flow of the results; on the other, it prevents a thorough and appropriately structured explanation of the analytical approaches and terminology. For example, I would appreciate a clearer explanation of the exploratory factor analysis, a more detailed description of the hierarchical omega and how it is computed, same for the guttman error/indice and clarification of the meaning of the "2+1 model." I therefore suggest moving the description of the analyses to the Methods section and including more detail to enhance transparency and reproducibility.

Thank you for this suggestion. Further detail on the factor analysis has been added to the manuscript in a new section (p 12-14 lines 292-324). This section clarifies what is meant by the 2+1 bi-factor model and also points towards R script which allows researchers follow and reproduce our methods.

Explanation of the omega coefficient and Guttman error index were no longer necessary within the main article, since the results of the response validity analysis had been condensed in this version for greater clarity (see p. 17-18 lines 383-396). Nevertheless, a more detailed explanation of these terms and the corresponding analysis can be found in the S3 appendix.

6. The final 8 items were selected from an initial pool of 45 items completed by the children. As the authors note, the results may differ in future assessments where only the 8-item version is administered. Therefore, I would like to see a comparison between the outcomes obtained using the full version and those from the final, shortened version. Did any participants in Study 2b complete the short version? If so, please compare their results with those who completed the full version. If not, it may be worthwhile to collect a small dataset—perhaps restricted to a specific grade level—and compare it to the results from the larger cohort that completed the full version. Doing so would substantially strengthen the validity of the proposed tool.

We agreed that it was necessary to test the 8-item scale in a new sample. Study 2 has been added to the manuscript which includes a sample of children aged 6-11 (N=56) who completed this final 8-item version of the cMI. We found that reliability estimates were similar to those gathered from the large sample recruited for study 1c. We also found that children’s responses were, on average, less random/inconsistent and extreme in comparison to those collected in study 1a.

7. In Study *1b, the thresholds assigned to the correlations appear somewhat arbitrary. I recommend including the correlation analyses and clearly defining the expected outcomes in the Methods section to ensure transparency and reproducibility.

The choice of effect size thresholds has been clarified in the Analysis section for study 1b (p. 22 lines 504-506). We have also moved the paragraph outlining expected outcomes into this Analysis section and added some further detail to justify our hypotheses.

8. According to the initial analyses, the authors explicitly acknowledge that the assessment may not be valid for the extreme age groups (6 and 13 years old). However, in Study *1c, they still collected data across the entire age range, and in the general discussion, they state that the test is suitable for ages 6 to 13. This appears somewhat inconsistent and would benefit from further clarification.

As has now been explained in p. 18 lines 387-393, the current results indicate that overall children in these age groups are more likely to provide random and extreme responses to the scale than 8–to 12-year-old children. This doesn’t, however, indicate that the measure is not valid for these groups overall.

To address your point, we have included further detail on recommended use of the scale in lines p. 34 lines 680-683 and p.36 lines 742-744. We make recommendations for reducing random/extreme responding by ensuring that 6- and 7-year-olds receive one-to-one supervision. We also recommend that 13-year-olds complete the GoldMSI as well as the cMI to ensure that researchers are able to capture the most of the meaningful variance in musicality/musical sophistication in this age group.

This will allow researchers’ to make an informed decision about their use of the scale with these age groups.

9. line 704 page 32 “versus”?

Removed.

10. Please include a description of the reliability assessment, as well as the alpha and omega parameters computed, in the Methods section for greater clarity and transparency.

An analysis section has been added to study 1c (p.28 lines 568-572) with the requested details.

Reviewer 2 comments:

The ms has an unfinished feel to it, and unless I'm missing something, we don't have a test of the actual proposed 8-item scale that includes only children who took the final scale.

We have addressed this concern by administering the final scale with a new sample. The new ‘study 2’ demonstrates the psychometric properties of the new scale in this sample.

Children from Study *1a seem to be included in all of the other studies.

Some data have been used in both study 1a and study 1b in order to investigate the validity and reliability of the task. To make it easier for a reader to establish that the same data have been used, we have amended study titles (i.e. studies 1, 2a, 2b and 2c are now 1a, 1b and 1c). A new study 2 reports insights from a separate sample.

The General Discussion is also particularly revealing--2 pages of a 38-page manuscript that doesn't convince me that I should ever use the scale, or that the authors even think that it's particularly useful.

We have combined the discussion sections across studies to enhance clarity and better promote the merits of the scale. The discussion now provides an overview of findings from both studies, including reports on validity and reliability, recommendations for use of the cMI and how it is expected to benefit researchers.

The results also go on and on and on, making the ms a chore to read.

To address this we have (1) combined the first two studies, previously 1 and 2a, and (2) condensed study 1c results. This has ensured that results sections include only the most critical information. We have also added analyses sections to each study, as recommended by reviewer 1, which has improved clarity of reporting and ensured the results sections are more focused.

As for the rationale, it's probably true that objective tests miss out on some aspects of musicality in childhood, but the authors assert this without actually convincing the reader why objective tests are a problem.

We do not argue that objective tests are a problem, instead we point out that they are only capturing a part of the meaningful variance in musicality during childhood (p. 5, lines 183-187). The cMI aims to identify and assess other interesting aspects of musicality that aren’t being captured by ability tests.

I note that in Table 1 and Figure 1, 2 of 3 items in the Music Making subscale have higher correlations with the General Musicality score than they do with the actual subscale. So why is Music Making a subscale? Similarly, for Musical Drive, 1 of 3 items is more strongly correlated with General Musicality than it is with Musical Drive, and another item has virtually identical correlations with the aggregate and subscale scores. In other words, the model of musicality embodied by the scale doesn't seem to work. Or at least the hierarchy of levels is not consistent with Carroll.

The high loadings of items onto both the general factor and group factors evidence their relevance to both and therefore support the bi-factor model we have proposed. This factor structure was identified as the best fitting structure according to the analysis which has been outlined in detail in the manuscript.

Note that in a bi-factor model the variance of each item is explained by the contribution of a general factor that is common to all items as well as by the contribution from a group factor that only comprises items from a subset of items. Ther

---

## [Decision Letter · Decision Letter 1]

4 Dec 2025

The child Musicality Index: a child-friendly version of the Goldsmiths Musical Sophistication Index

PONE-D-25-11934R1

Dear Dr. MacGregor,

We’re pleased to inform you that your manuscript has been judged scientifically suitable for publication and will be formally accepted for publication once it meets all outstanding technical requirements.

Kind regards,

Gal Harpaz, Ph.D.

Academic Editor

PLOS ONE

Additional Editor Comments (optional):

Reviewers' comments:

Reviewer's Responses to Questions

**Comments to the Author**

Reviewer #1: All comments have been addressed

2. Is the manuscript technically sound, and do the data support the conclusions?

Reviewer #1: Yes

3. Has the statistical analysis been performed appropriately and rigorously?

Reviewer #1: Yes

4. Have the authors made all data underlying the findings in their manuscript fully available?

Reviewer #1: Yes

5. Is the manuscript presented in an intelligible fashion and written in standard English?

Reviewer #1: Yes

Reviewer #1: (No Response)

**Do you want your identity to be public for this peer review?** For information about this choice, including consent withdrawal, please see our Privacy Policy

Reviewer #1: No

---

## [Editor Report · Acceptance letter]

PONE-D-25-11934R1

PLOS One

Dear Dr. MacGregor,

I'm pleased to inform you that your manuscript has been deemed suitable for publication in PLOS One. Congratulations! Your manuscript is now being handed over to our production team.

Kind regards,

on behalf of

Dr. Gal Harpaz

Academic Editor

PLOS One